# Cerebrospinal Fluid and Peripheral Blood Lymphomonocyte Single-Cell Transcriptomics in a Subject with Multiple Sclerosis Acutely Infected with HIV

**DOI:** 10.3390/ijms251910459

**Published:** 2024-09-28

**Authors:** Carmela Pinnetti, Gabriella Rozera, Francesco Messina, Pietro Giorgio Spezia, Elisabetta Lazzari, Lavinia Fabeni, Giovanni Chillemi, Daniele Pietrucci, Shalom Haggiag, Ilaria Mastrorosa, Alessandra Vergori, Enrico Girardi, Andrea Antinori, Fabrizio Maggi, Isabella Abbate

**Affiliations:** 1Clinical and Research Infectious Department, National Institute for Infectious Diseases Lazzaro Spallanzani-IRCCS, 00149 Rome, Italy; carmela.pinnetti@inmi.it (C.P.); ilaria.mastrorosa@inmi.it (I.M.); alessandra.vergori@inmi.it (A.V.); andrea.antinori@inmi.it (A.A.); 2Laboratory of Virology, National Institute for Infectious Diseases Lazzaro Spallanzani-IRCCS, 00149 Rome, Italy; pietro.spezia@inmi.it (P.G.S.); elisabetta.lazzari@inmi.it (E.L.); lavinia.fabeni@inmi.it (L.F.); fabrizio.maggi@inmi.it (F.M.); isabella.abbate@inmi.it (I.A.); 3Laboratory of Microbiology, National Institute for Infectious Diseases Lazzaro Spallanzani-IRCCS, 00149 Rome, Italy; francesco.messina@inmi.it; 4Department for Innovation in Biological, Agro-Food and Forest Systems (DIBAF), University of Tuscia, 01100 Viterbo, Italy; gchillemi@unitus.it (G.C.); daniele.pietrucci@unitus.it (D.P.); 5Neurology Department, San Camillo Forlanini Hospital, 00152 Rome, Italy; shaggiag@scamilloforlanini.rm.it; 6Scientific Direction, National Institute for Infectious Diseases Lazzaro Spallanzani-IRCCS, 00149 Rome, Italy; enrico.girardi@inmi.it

**Keywords:** interactions between infective agents and immune responses, immune dysregulation in infectious diseases

## Abstract

Signatures of neurodegeneration in clinical samples from a subject with multiple sclerosis (MS) acutely infected with HIV were investigated with single-cell transcriptomics using 10X Chromium technology. Sequencing was carried out on NovaSeq-TM, and the analysis was performed with Cell Ranger software (v 7.1.0) associated with a specifically established bioinformatic pipeline. A total of 1446 single-cell transcriptomes in cerebrospinal fluid (CSF) and 4647 in peripheral blood mononuclear cells (PBMCs) were obtained. In the CSF, many T-cell lymphocytes with an enriched amount of plasma cells and plasmacytoid dendritic (pDC) cells, as compared to the PBMCs, were detected. An unsupervised cluster analysis, putting together our patient transcriptomes with those of a publicly available MS scRNA-seq dataset, showed up-regulated microglial neurodegenerative gene expression in four clusters, two of which included our subject’s transcriptomes. A few HIV-1 transcripts were found only in the CD4 central memory T-cells of the CSF compartment, mapping to the *gag-pol*, *vpu*, and *env* regions. Our data, which describe the signs of neurodegenerative gene expression in a very peculiar clinical situation, did not distinguish the cause between multiple sclerosis and HIV infection, but they can give a glimpse of the high degree of resolution that may be obtained by the single-cell transcriptomic approach.

## 1. Introduction

HIV invades the central nervous system (CNS) of infected subjects soon after infection, where it can persist, evolve, and become compartmentalized [1]. Although the productive infection in this compartment is low, the presence of the virus, together with local inflammation, may lead to neurodegenerative disorders [2].

Cerebrospinal fluid (CSF) is a routinely and safely accessible component of the CNS. It constitutes a unique local immune medium that can be used to diagnose inflammatory and degenerative diseases affecting the CNS [3,4,5,6]. CSF cells derive exclusively from the hematopoietic lineage and exhibit a tightly regulated cellular composition that differs significantly from that of blood [7].

Single-cell transcriptomics can define the heterogeneity of the cellular immune response to different autologous and external stimuli. Analyzing the types of cells that show up in the CSF with their specific transcriptomes could clarify the events involved in the development of neuroinflammatory and neurodegenerative diseases in the CNS.

In multiple sclerosis (MS), the most common inflammatory demyelinating disease of the CNS, single-cell transcriptomics has already highlighted an increased transcriptional diversity in the blood, together with an increased cell type diversity in the CSF compared to healthy donors. This suggests a compartmentalized mechanism leading to autoimmunity in the brain [8].

Previous studies of the CSF during HIV infection have primarily focused on the soluble immune markers of inflammation [9,10,11,12]. More recently, by applying a single-cell transcriptomic experimental approach, a subset of myeloid cells was identified in the CSF of infected subjects, expressing a transcriptomic signature similar to that found in neurodegenerative disease-associated microglia [13,14].

The present study aimed to establish single-cell transcriptomes in the CSF and peripheral blood mononuclear cells (PBMCs) in a person diagnosed with an HIV primary infection (PHI), naïve to antiretroviral therapy and already affected by MS, to characterize the possible signatures of neurodegeneration and to find viral transcripts in the examined cells.

## 2. Results

This study was conducted on a person with MS who was developing a primary infection with HIV-1. At the time, a lumbar puncture (LP) was performed, and a CSF examination revealed a protein concentration of 70.5 mg/dL (normal values range from 8 to 32 mg/dL). The CSF and plasma albumin were 4.6 mg/L (normal values range from 3.5 to 5.1 mg/L) and 4250 mg/dL (normal values range from 3200 to 4800 mg/dL), respectively. We did not observe any evidence of blood–brain barrier damage. All microbiological tests, including rapid plasma reagin (RPR) and polymerase chain reaction (PCR), were negative for *Treponema pallidum*. The HIV RNA levels in the CSF/plasma pair were 7195/69,396 copies/mL and his CD4 T-cell count was 323/mm^3^ (29.9%), with a CD4/CD8 ratio of 0.63. Other blood chemistry tests were within normal limits. The HIV *gag*, pol, and *env* sequencing identified an HIV-1 G subtype (submitted to NCBI’s BankIt platform Accession numbers: PQ287393, PQ287392, and PQ287394, respectively, and will be available from 3 January 2025). Single-cell whole transcriptomes, from high-quality cDNA, produced a mean of 1594 transcripts/cell in the CSF and 6255 transcripts/cell in the PBMCs, with a low proportion of mitochondrial genes (<5%). A total of 1446 single-cell transcriptomes were obtained from the CSF, and 4647 from the PBMCs. To identify classical cell clusters in the separated diffusion maps obtained from the transcriptomes of the two clinical samples and, therefore, to characterize the whole compartment-specific composition of the CSF and PBMCs, we used the same canonical marker genes as described in the Section 4 (Figure 1A,B).

The analysis indicated that most cells (85.5%) in the CSF were T lymphocytes, with an enriched amount of plasma cells and plasmacytoid dendritic (pDC) cells, compared to PBMCs. Conversely, B cells and natural killer (NK) cells were less represented in the CSF than in PBMCs (Table 1).

Then, to highlight potential rare cells with specific transcriptomic signatures of neurodegeneration, in this case, an unsupervised cluster analysis was performed. This analysis did not rely on known cell type markers, but instead combined the patient’s single-cell transcriptomes, derived from the CSF and PBMC clinical samples, with those of publicly available scRNA-seq data. The resulting diffusion map (Figure 2) identified 16 cell clusters with significant differential gene expression (clusters 0–15). Clusters 6, 12, 14, and 15 exhibited up-regulated gene expression, previously associated with a microglial neurodegenerative phenotype, as described in Farhadian et al. [13] (see arrows).

In these clusters, gene expression analysis revealed the presence of several cell-specific markers. Among these, the *Macrophage Scavenger Receptor 1* (*MSR-1*) was identified in cluster 6; the myeloid dendritic (mDC) marker high-affinity IgE Receptor (FcεRI) emerged in clusters 12 and 14; the generic myeloid marker CD33 was detected in clusters 14 and 15; and non-classical resident monocyte markers as the *CX3C motif chemokine receptor 1* (*CX3CR1*), the *Macrophage Activating Factor* (*MAF*), and the *Colony Stimulating Factor 1 Receptor* (*CSF1R*) were identified in cluster 15. The results indicated that some of the patient’s single-cell transcriptomes were included in both cluster 6 (2.3% of the total CSF and 10.4% of the total PBMC patient-specific transcriptomes) and cluster 14 (0.02% of total the PBMC patient-specific transcriptomes). Among the microglial neurodegenerative phenotype-associated gene expression, we observed in cluster 6 an upregulation of transcripts coding for several proteins, including complement C1q subcomponent subunits A (*C1QA*), B (*C1QB*), and C (*C1QC*), the class II major histocompatibility complex-associated protein CD74, cathepsin L, ferritin light chain (FTL), and both the *Macrophage Receptor with Collagenous Structure* (*MARCO*) and *MSR-1*. In cluster 14, we observed upregulated transcripts for *Apolipoprotein C1* (*APOC1*), the AXL receptor tyrosine kinase, Human Leukocyte Antigen (HLA) molecules, the *IL18* gene, the *Solute Carrier Family 25 Member 5* (*SLC25A5*), which is involved in RNA binding and adenine transmembrane transport, and the *Parathymosin* gene (*PTMS*). Additionally, clusters 6 and 14 shared the upregulation of three genes related to neurodegenerative patterns, such as the transcript coding for Dihydropyrimidinase Like 2 (DPYSL2), Thymosin Beta 10 (TMSB), and Vesicle Associated Membrane Protein 8 (VAMP8). The complete list of up-regulated gene expressions for each cluster, shown in Figure 2, is included in Appendix A.

A few HIV-1 transcripts (a total of eleven) mapping to the *gag*, pol, vpu, and env regions were detected only in the CSF compartment, specifically within three CD4^+^ central memory T-cells (Figure 3).

## 3. Discussion

Although limited to a single individual with a very specific clinical situation, i.e., a patient affected by MS with a superimposed primary HIV infection, our findings provide a glimpse into the potential of a single-cell transcriptomic experimental approach for understanding pathogenetic events at the single-cell level in a complex disease scenario.

First, with unbiased cell type characterization, the cell types present in the CSF of the MS/HIV-1-infected patient were successfully defined. This finding confirms a compartment-specific cell composition of the CSF, consistent with previous studies [8,14]. Notably, there is a particular enrichment of plasma cells concerning peripheral blood, a phenomenon already shown in other MS patients [15].

The second result is the identification of the early signs of neurodegeneration in the myeloid cells. Specifically, some patients’ myeloid cells share a specific transcriptomic profile with those from patients with neurodegenerative disorders previously described in other studies [8]. A particularly significant finding is that the patient’s cell transcriptomes, represented in cluster 6 of Figure 2, display an upregulated expression of the MSR-1 gene, a well-known hallmark of activated microglia and CNS perivascular macrophages [14]. However, we cannot exclude the possibility of underestimating the complete cellular transcription landscape, potentially linked to neurodegenerative phenotypes, due to the relatively low number of analyzed transcriptomes. This limitation arises from analyzing data from just one subject and the difficulty of finding similar cases with the same clinical characteristics. Finally, HIV transcripts within the cellular transcripts from the analyzed CSF and PBMC samples were detected. These were found only in a few CD4 T central memory cells in the CSF compartment, but not in cells of myeloid origin. The cellular-associated HIV transcripts detected in the CSF compared to the plasma are consistent with findings reported by Farhadian et al. [14], although their study involved HIV patients undergoing antiretroviral therapy. This discrepancy can also be attributed to a relative enrichment of infected cells within the CSF compartment as compared to the total leukocyte population. HIV infection in leukocytes has been shown to enhance their transmigration across the blood–brain barrier (BBB), a process facilitated by soluble inflammatory factors, including Chemokine (C-C motif) Ligand 2 (CCL2), which is also a key mediator in the pathogenesis of multiple sclerosis [16,17]. One limitation of our study is the lack of a simultaneous immunological characterization of the CSF sample, due to the need to use a large volume of CSF for virologic analysis. Additionally, the non-specific nature of the observed signs of myeloid neurodegeneration makes it difficult to determine the relative contribution of multiple sclerosis and the primary HIV infection as the causative agents of the observed condition.

In conclusion, this study, which addresses a peculiar clinical case of primary HIV infection in a subject already affected by MS, demonstrates the power of single-cell transcriptomic technology to either characterize, using an unbiased approach, the cellular composition of the CSF and to appreciate the initial signs of the neurodegenerative phenotype in myeloid cells. However, efforts are needed to improve the sensitivity of the detection of viral transcripts among host cellular transcriptomes to better evaluate direct host–pathogen interactions.

## 4. Materials and Methods

### 4.1. Clinical and Virological Patient Characteristics

A 37-year-old individual with a prior diagnosis of MS acutely infected with HIV was studied. The subject was diagnosed with MS twelve years earlier and underwent treatment with γ-interferon and steroid therapy for the first five years, experiencing alternating phases of remission and recrudescence. The subject was then lost to follow-up. Approximately ten years prior, a diagnosis of syphilis was reported, which was treated with intramuscular benzyl penicillin. The subject returned to the neurology ward about a year ago due to a recurrence of neurological symptoms with a gradual worsening of hyposthenia in the left lower limb. A magnetic resonance imaging (MRI) scan of the brain and cervical spine was conducted to reassess the disease status (see Appendix A). During screening examinations prior to potential administration of MS immune-modulating therapy, the patient tested positive for HIV (a previous negative test was reported about a year earlier). The diagnosis of PHI was based on a low avidity index of the anti-HIV antibodies and clinical evaluation. The subject reported engaging in unprotected intercourse with a partner who had been previously diagnosed with HIV and was on antiretroviral therapy with poor adherence. The HIV-1 RNA levels in the plasma and CSF were quantified by a commercial assay (Aptima™ HIV-1 Quant Dx Assay, Hologic, Bedford, MA, USA). Molecular characterization of the HIV strain was performed by PCR amplification of three regions of the viral genome (*gag*, *env*, and *pol*) as described in the literature [18,19]. HIV subtyping was performed using REGA software available on the Genome Detective platform, version 3.46 (https://www.genomedetective.com/app/typingtool/hiv/, accessed on 25 July 2024). The patient started combination antiretroviral therapy with tenofovir, emtricitabine, and dolutegravir only after undergoing lumbar puncture. The therapy is ongoing and well tolerated. The study participant provided informed consent.

### 4.2. Single-Cell Transcriptomic Libraries and Sequencing

For single-cell transcriptomics, 9 mL of CSF obtained by lumbar puncture was centrifuged, and the cellular pellet, in parallel with PBMCs collected from peripheral blood by centrifugation on the Fycoll gradient, was washed before cell counting. The viable cells were measured using trypan blue. Approximately, 5000 cells per sample were processed to reverse transcription and barcoding using 10X Chromium Single Cell reagents and platform (10X Genomics, Pleasanton, CA, USA). Each single barcoded cell generated a cDNA library according to the Single Cell 3′ Reagent Kits v3 protocol (10X Genomics) and was fragmented and ligated to adapters for sequencing on the Illumina platform NovaSeqTM 6000 System (Illumina, San Diego, CA, USA), using at least 20,000 read pairs per single barcoded cell.

### 4.3. Bioinformatic Analysis

The bioinformatic analysis was conducted with the Cell Ranger software (version 7.1.0) [20], which analyzed the transcripts derived from each single cell. Subsequently, a clustering analysis was performed, and the gene expression was compared among the clusters. LogFC > |0.1| e *p* < 0.05 were considered statistically significant. Canonical markers were used to assign the identity of the cell clusters, as in [8]. To increase the likelihood of detecting rare cells displaying transcriptomic signatures of neurodegenerative disease, a publicly available scRNA-seq dataset (GEO accession code: GSE141797) was retrieved from [8] and included in a second cluster analysis.

The presence of HIV-1 transcripts was investigated by a specifically established bioinformatic pipeline. Briefly, to detect HIV viral transcripts in 10X Genomics scRNA-Seq data from blood and CSF, a complete consensus genome sequence of HIV-1 subtype G (NCBI nucleotide code: KU168271) was utilized to generate a custom reference with the CellRanger mkref function within the 10X Genomics Cell Ranger pipeline. FASTA and the GFF3 files were acquired from NCBI. Since the CellRanger mkref pipeline requires FASTA sequences and GTF files, the gffreads utility from Cufflinks (version 2.2.1) was used to convert the GFF3 file into GTF format [21]. Subsequently, reads mapped to the reference genome were identified using Samtools (version 1.13) [22], and their alignment with the reference HIV genome was visualized using the Integrative Genomics Viewer (IGV) Web App [23]. To identify the cells containing HIV transcripts, the barcodes of the cells associated with the retrieved transcripts were used. The analysis was conducted using the statistical software R (version 4.3.2) along with the following packages: Seurat v.3.0.2 [24], SingleR v.2.4.0, and celldex v.1.12.0 [25]. The data analysis followed a procedure similar to that reported by Léon-Rivera et al. [26]. The dataset was filtered using the Seurat package to remove all cells that did not express at least 200 genes and contained less than 25% mitochondrial DNA. In total, 65,237 cells were analyzed. Subsequently, dimensionality reduction and clustering analysis were performed using UMAP [27]. Clustering was carried out using the default graph-based Louvain algorithm, and the FindClusters function was used with a resolution range of 0.5. The cell type inference for each cluster was then performed using the SingleR package, utilizing the “PBCM” collection from the celldex package.

## Figures and Tables

**Figure 1 ijms-25-10459-f001:**
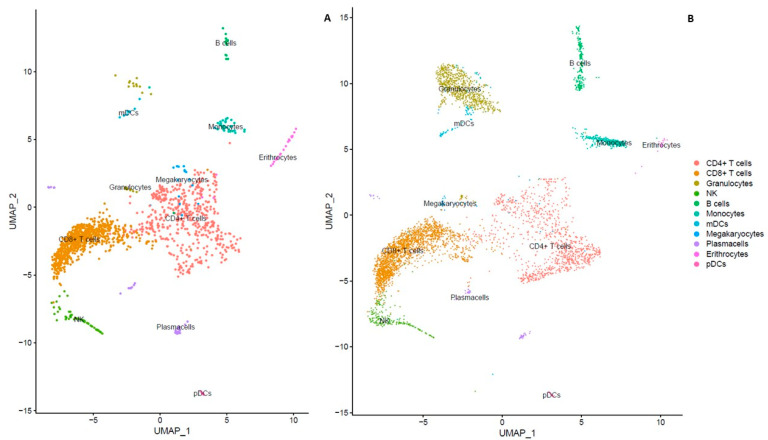
Uniform manifold approximation and projection (UMAP) plots representing cell clusters identified in patient’s CSF and PBMC single-cell transcriptomes. Color-coded cell clusters derived from single-cell transcriptomic analysis of CSF (**A**) and PBMC (**B**) samples are shown. To assess the cellular identity of each cluster, canonical cell markers were used (see Section 4).

**Figure 2 ijms-25-10459-f002:**
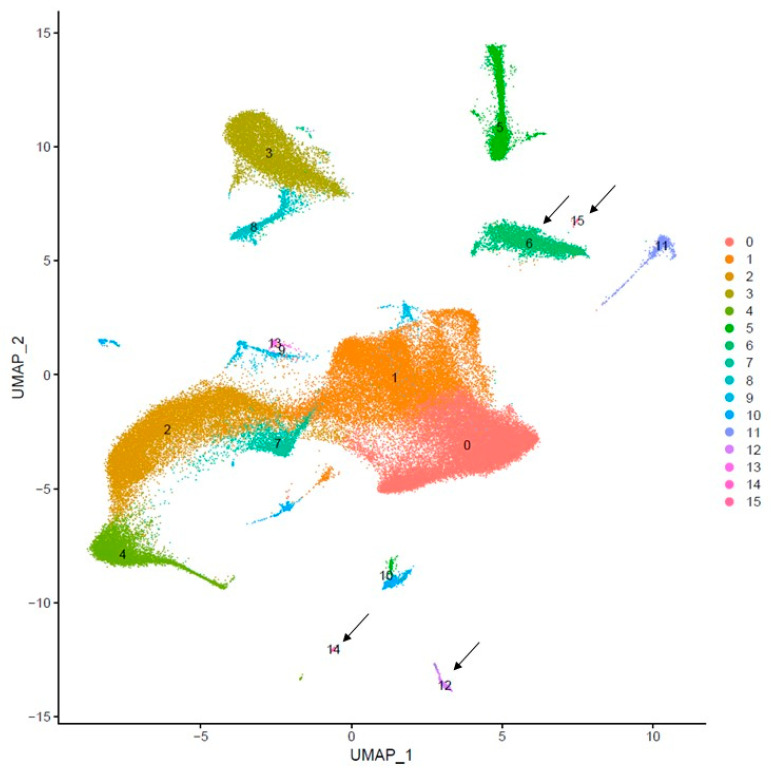
Uniform manifold approximation and projection (UMAP) plots representing 0–15 clusters identified in unsupervised cluster analysis performed on the total CSF and PBMC transcriptomes. The 16 identified clusters are shown using different colors (see legend on the right) and the arrows indicate those displaying up-regulated neurodegenerative microglial gene expression.

**Figure 3 ijms-25-10459-f003:**
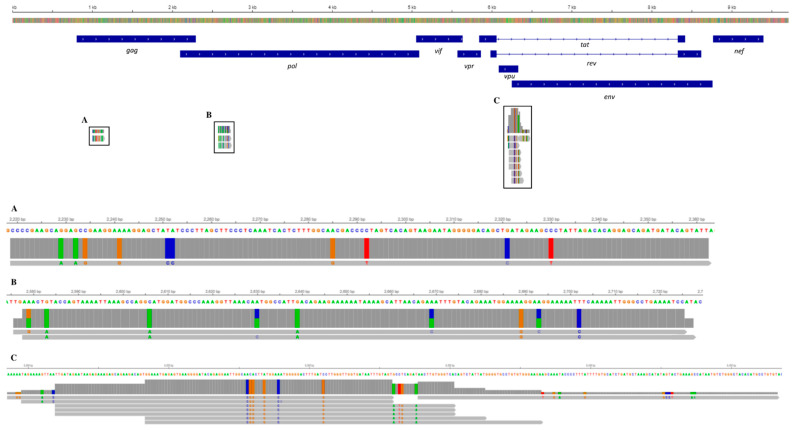
Alignment track of HIV transcripts in single cells from the CSF across the annotated HIV subtype G (KU168277). On the top, the HIV reference sequence and annotated genes (in dark blue) are displayed. (A–C): the read coverage and stacked individual reads, represented as arrows, are shown in different panels enlarged below. Different colors within the reads highlight present nucleotide variations compared to the reference sequence: red is used for T, green is used for A, orange is used for G, and blue for C. Picture obtained from the IGV Web App.

**Table 1 ijms-25-10459-t001:** Cell type frequencies in the CSF and PBMCs. The identification of different cell types in the CSF and PBMCs was performed using canonical marker genes as shown in Figure 1.

Cell Type	CSF (%)	PBMC (%)
CD4^+^ T-Cells	32.92	20.46
CD8^+^ T-Cells	52.63	42.31
Granulocytes	1.73	15.73
NK	3.60	4.41
B Cells	0.90	4.26
Monocytes	2.28	9.12
mDCs	0.55	1.08
Megakaryocytes	1.04	0.67
Plasma cells	2.21	1.25
Erythrocytes	1.73	0.47
pDCs	0.41	0.24

## Data Availability

Dataset is available on request from the authors. The raw data supporting the conclusions of this article will be made available by the authors on request.

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
