# Peer review of "Cerebrospinal Fluid and Peripheral Blood Lymphomonocyte Single-Cell Transcriptomics in a Subject with Multiple Sclerosis Acutely Infected with HIV"

_ijms, 2024, doi:10.3390/ijms251910459_

Round 1

Reviewer 1 Report

Comments and Suggestions for Authors

The paper is well written, but use of some terminology like "district" in abstract is a bit confusing.  "Compartment" would be a better and accurate description.  Also such analyses with multiple patients with HIV on ART, and with nascent HIV infection have been described elsewhere.  Furthermore, the subset of cells with MSR1 has also been documented and described elsewhere.  That observation is not novel.  

The novelty in this study is having a sample with MS and HIV.  But this being a report on a single sample does not have the statistical power to draw meaningful conclusions.

Comments on the Quality of English Language

English requires significant editing efforts.  A lot of terminology is terse and poorly described.  

Author Response

Rome, September 13, 2024

Point-by-point response letter:

Reviewer #1:

The paper is well written, but use of some terminology like "district" in abstract is a bit confusing.  "Compartment" would be a better and accurate description.  Also, such analyses with multiple patients with HIV on ART, and with nascent HIV infection have been described elsewhere.  Furthermore, the subset of cells with MSR1 has also been documented and described elsewhere.  That observation is not novel. 

The novelty in this study is having a sample with MS and HIV.  But this being a report on a single sample does not have the statistical power to draw meaningful conclusions.

Comments on the Quality of English Language

English requires significant editing efforts.  A lot of terminology is terse and poorly described. 

Reply: We replace the term “district” with “compartment” throughout the manuscript.

We better described some used terminology such as the acronyms are now spelled out, please see lines: 72, 83, 106, 128, 130 - 134, and 139 – 146.

The new version has been reviewed by an English text editor, with experience in medical writing.

Reviewer 2 Report

Comments and Suggestions for Authors

This manuscript from Pinnetti C. et al. underscores the significance of RNA single-cell transcriptome analysis, a widely recognized and powerful technique, as emphasized even in its conclusions.

The manuscript presents several major considerations that, in this reviewer's opinion, warrant correction or modification:

As a record, have the authors any data of magnetic resonance imaging (MRI) findings in this case? Beyond the syphilis serological test, have you considered that in neurosyphilis typically reveals vasculitis and infarction, aiding in distinguishing it from MS? Is this patient under antiretroviral therapy?

Line 82, page 2: "RPR" clarify

Line 83, page 2: How the authors explain the different VL levels among CSF and plasma (9-fold change in CSF)? Could this VL related to the number of HIV transcripts characterized in three T lymphocytes? The authors have characterized HIV-1 transcripts in the CSF compartment, "specifically within three CD4+ central memory T cells" but not among blood cells.

Line 85, page2: "HIV gag, pol, and env sequencing assigned an HIV-1 G subtype" Have the authors deposited these sequences in a database? Please indicate the numbers to access. 

Line 90: When the authors refer "to characterize the whole compartment-specific composition of the CSF and PBMC" are studying CSF and blood (not PBMC) compartments.  Specify clearly the total number of mononuclear cells that they counted among each. 

Have the authors tried to study comparatively their own data with other already published in patients with HIV infection, or MS separately? This analysis with a heat map would be very useful for the analysis.

"Table 1: Cell type frequencies in CSF and PBMC". The data are not only about mononuclear cells because also granulocytes (and erytrocytes) are characterized. Hence, these are cells from CSF and blood.

Caption of Figure 3: "Different colors within the reads highlight present nucleotide variations compared to the 151 reference sequence" It is really hard to differentiate colours. Enlarge or zoom.  

-------------------------------------------------------------------------------------

Minor comment:  Check the "PBMC" term throughout the whole manuscritp.

Author Response

Rome, September 13, 2024

Reviewer #2:

This manuscript from Pinnetti C. et al. underscores the significance of RNA single-cell transcriptome analysis, a widely recognized and powerful technique, as emphasized even in its conclusions.

The manuscript presents several major considerations that, in this reviewer's opinion, warrant correction or modification:

As a record, have the authors any data of magnetic resonance imaging (MRI) findings in this case? Beyond the syphilis serological test, have you considered that in neurosyphilis typically reveals vasculitis and infarction, aiding in distinguishing it from MS? Is this patient under antiretroviral therapy?

Reply: We included the record of the magnetic resonance imaging (MRI) of the brain and cervical spine of the study patient as supplemental material (Figure S1, see at the end of the manuscript attached) lines 216-217.

The patient has never exhibited neurosyphilis. The diagnosis of secondary syphilis was made several years earlier (the HIV test at that time was negative) and was treated with intramuscular benzathine penicillin (as specified at lines 213-214). At the time of the HIV diagnosis, serological tests were negative (as described at lines 82-84). The lesions documented by MRI were already present in the past and had characteristics compatible with multiple sclerosis lesions.

The patient started combination antiretroviral therapy with tenofovir, emtricitabine, and dolutegravir only after undergoing a lumbar puncture. The therapy is ongoing and well tolerated. We specified this by adding the following sentence at lines 228-231.

At the time of evaluation, the patient was naïve to antiretroviral therapy. We included this information in the manuscript revised version at line 73.

Line 82, page 2: "RPR" clarify

Reply: "RPR" represents the abbreviation of “Rapid Plasma Reagine” and is a non-treponemal serologic tests used to screen for active syphilis infection and to monitor response to treatment. We specified this abbreviation in the revised version of the manuscript, please refer to line 83.

Line 83, page 2: How the authors explain the different VL levels among CSF and plasma (9-fold change in CSF)? Could this VL related to the number of HIV transcripts characterized in three T lymphocytes? The authors have characterized HIV-1 transcripts in the CSF compartment, "specifically within three CD4+ central memory T cells" but not among blood cells.

Reply: We apologize, but the HIV-1 RNA levels have been inverted in the text for a typing mistake. We now corrected in the revised version of the paper that “HIV RNA in the CSF/plasma pair was 7,195/69,396 copies/ml.”.

The cellular-associated HIV transcripts detected in cerebrospinal fluid (CSF) compared to plasma is consistent with findings reported by Farhadian et al. (2002), although their study involved HIV patients undergoing antiretroviral therapy. This discrepancy can also be attributed to a relative enrichment of infected cells within the CSF compartment as compared to the total leukocyte population. HIV infection in leukocytes has been shown to enhance their transmigration across the blood-brain barrier (BBB), a process facilitated by soluble inflammatory factors, including the Chemokine (C-C motif) Ligand 2 (CCL2) (Eugenin et al.), which is also a key mediator in the pathogenesis of multiple sclerosis (Cheng et al.).

We added this consideration in the discussion (see lines 185- 193) we included the relative 2 references (now 16 and 17 respectively) listed below:

Eugenin, E.A.; Osiecki, K.; Lopez, L.; Goldstein, H.; Calderon, T.M.; Berman, J.W. CCL2/Monocyte Chemoattractant Protein-1 Mediates Enhanced Transmigration of Human Immunodeficiency Virus (HIV)-Infected Leukocytes across the Blood–Brain Barrier: A Potential Mechanism of HIV–CNS Invasion and NeuroAIDS. J. Neurosci. 2006, 26, 1098–1106, doi:10.1523/JNEUROSCI.3863-05.2006.

Cheng, W.; Chen, G. Chemokines and Chemokine Receptors in Multiple Sclerosis. Mediators of Inflammation 2014, 2014, 659206, doi:10.1155/2014/659206.

Line 85, page2: "HIV gag, pol, and env sequencing assigned an HIV-1 G subtype" Have the authors deposited these sequences in a database? Please indicate the numbers to access. 

Reply: HIV gag, pol and env sequences were submitted to GenBank’s platform BankIt, and will be available from March 1st in 2025 and the respective accession numbers are: PQ287393, PQ287392 and PQ287394 respectively (lines 87-88).

Line 90: When the authors refer "to characterize the whole compartment-specific composition of the CSF and PBMC" are studying CSF and blood (not PBMC) compartments.  Specify clearly the total number of mononuclear cells that they counted among each. 

Reply: The characterization was made by analyzing the results of single cell transcriptomics, and for this analysis we used counted 5,000 patient’s cells deriving either from the cellular pellet of CSF or from PBMC collected on the Fycoll gradient.

Have the authors tried to study comparatively their own data with other already published in patients with HIV infection, or MS separately? This analysis with a heat map would be very useful for the analysis.

Thank you for your suggestion. We have compared our gene expression data from CSF and PBMCs with a publicly available scRNA-seq dataset (GEO accession code: GSE141797 Schafflick et al.,2020:10.1038/s41467-019-14118-w), reporting gene expression data analysis on CSF leukocytes from patients with multiple sclerosis and healthy controls. This comparison was conducted to enhance the detection of rare cells exhibiting transcriptomic signatures associated with neurodegenerative disease and to refine the cluster analysis of cell types. We also tried to perform the analysis using Farhadian et al. data but, the reduced sample size and the different methodology employed by the authors prevented us from achieving suitable clustering results. However, further comparisons with other datasets from HIV and multiple sclerosis patients could be misleading due to significant inter-patient variability (in our case, involving only a single patient) and potential batch effects from differing experimental conditions.

"Table 1: Cell type frequencies in CSF and PBMC". The data are not only about mononuclear cells because also granulocytes (and erytrocytes) are characterized. Hence, these are cells from CSF and blood.

Reply: As stated before, the characterization was made by analyzing the results of single-cell transcriptomics. The proportions of granulocytes and erythrocytes, shown in Table 1, especially for PBMC, are probably due to residual cell contamination that occurred during the gradient purification process.

Caption of Figure 3: "Different colors within the reads highlight present nucleotide variations compared to the reference sequence" It is really hard to differentiate colours. Enlarge or zoom.  

Reply: Figure 3 has been modified as requested.

Minor comment:  Check the "PBMC" term throughout the whole manuscript.

Reply: Ok, the term PBMC was checked throughout the whole manuscript.

Reviewer 3 Report

Comments and Suggestions for Authors

This brief report provides data on a single patient who was recently infected with HIV but was previously diagnosed with MS.  The data consists of scRNA-seq data on cells isolates from the CSF and PBMC at a single timepoint.  The authors discuss potential signatures of neurodegeneration and viral transcripts present is some of these cell populations.  Although this is a limited dataset that doesn't necessarily have the highest impact, the dataset may be useful to researchers in the field.  The brief report is concise and easy to read.  Minor suggested changes below

Suggested Changes:

Line 30, italicize “ad hoc”.  It is italicized in the methods section

Line 49, reword “HIV early invades the central nervous system (CNS) of infected subjects” to “HIV invades the central nervous system (CNS) of infected subjects soon after infection” or something similar

Line 51, change “district” to “compartment”

Lines 79-81 “At this time, a lumbar puncture (LP) was performed and a CSF examination showed a protein concentration of 70.5 mg/dl. CSF and plasma albumin were 4.6 mg/l and 4250 mg/dl, respectively”, it may be helpful to the reader to list the normal ranges or state that all of these measurements fell within the normal range.

Line 102, clarify “concerning” in this context.  Maybe rewrite this sentence.  “in relation to PBMC”?  “compared to PBMC”?

Throughout the text, including the title, you may want to change the description of the subject “A 37-year-old person with a previous diagnosis of MS acquiring HIV infection was studied.  It reads as thought the acquisition of HIV is currently happening.  Perhaps change to “who recently acquired HIV”, “acutely infected”, or phrasing that indicates the infection has already happened.

Line 218 “was washed and viable counted using trypan blue”, change to “was washed before cell number and viability were measured using trypan blue”

Comments on the Quality of English Language

Please see suggested changes above.

Author Response

Rome, September 13, 2024

Reviewer #3:

This brief report provides data on a single patient who was recently infected with HIV but was previously diagnosed with MS.  The data consists of scRNA-seq data on cells isolates from the CSF and PBMC at a single timepoint.  The authors discuss potential signatures of neurodegeneration and viral transcripts present is some of these cell populations.  Although this is a limited dataset that doesn't necessarily have the highest impact, the dataset may be useful to researchers in the field.  The brief report is concise and easy to read.  Minor suggested changes below

Suggested Changes:

Line 30, italicize “ad hoc”.  It is italicized in the methods section

Reply: Ok, we changed it with “specifically established”, line 29 and 253.

Line 49, reword “HIV early invades the central nervous system (CNS) of infected subjects” to “HIV invades the central nervous system (CNS) of infected subjects soon after infection” or something similar

Reply: Ok, the phrase was changed as suggested, lines 48-49.

Line 51, change “district” to “compartment”

Reply: Ok, compartment instead of district was used, throughout the manuscript.

 Lines 79-81 “At this time, a lumbar puncture (LP) was performed and a CSF examination showed a protein concentration of 70.5 mg/dl. CSF and plasma albumin were 4.6 mg/l and 4250 mg/dl, respectively”, it may be helpful to the reader to list the normal ranges or state that all of these measurements fell within the normal range.

Reply: As requested we added the reference values, lines 79-81.

Line 102, clarify “concerning” in this context.  Maybe rewrite this sentence.  “in relation to PBMC”?  “compared to PBMC”?

Reply: OK it was changed in compared to PBMC.

 Throughout the text, including the title, you may want to change the description of the subject “A 37-year-old person with a previous diagnosis of MS acquiring HIV infection was studied.”  It reads as thought the acquisition of HIV is currently happening.  Perhaps change to “who recently acquired HIV”, “acutely infected”, or phrasing that indicates the infection has already happened.

Reply: OK it was changed in ‘acutely infected’ (lines 27 and 209) and ‘soon after infection’ (line 48).

Line 218 “was washed and viable counted using trypan blue”, change to “was washed before cell number and viability were measured using trypan blue”

Reply: OK the phrase was changed as suggested.

Round 2

Reviewer 2 Report

Comments and Suggestions for Authors

The comments made on the original version have been satisfactorily addressed and, where appropriate, have led to modifications in the manuscript.